# Identification of diabetic retinopathy lesions in fundus images by integrating CNN and vision mamba models

**Zenglei Liu**[1], **Ailian Gao**[1]*, **Hui Sheng**[2], **Xueling Wang**[2]*

**1** School of Electrical and Information Engineering, Hunan Institute of Technology, Hengyang, China,
**2** Department of Radiology, Yantaishan Hospital, Yantai, Shandong, China

* 13333775907@163.com (AG); m15315561072@163.com (XW)

## Abstract

Diabetic retinopathy, a retinal disorder resulting from diabetes mellitus, is a prominent cause of visual degradation and loss among the global population. Therefore, the identification and classification of diabetic retinopathy are of utmost importance in the clinical diagnosis and therapy. Currently, these duties are extensively carried out by manual examination utilizing the human visual system. Nevertheless, manual examination is sometimes arduous, time-consuming, and prone to errors. Deep learning-based methods have recently demonstrated encouraging results in several areas, such as image categorization and natural language mining. The majority of deep learning techniques developed for medical image analysis rely on convolutional modules to extract the inherent structure of images within a certain local receptive field. Furthermore, transformer-based models have been utilized to tackle medical image processing problems by capitalizing on global connections among distant pixels in the images. Considering these analyses, this work presents a comprehensive deep learning model that combines convolutional neural network and vision mamba models. This model is designed to accurately identify and classify diabetic retinopathy lesions displayed in fundus images. Furthermore, the vision mamba component incorporates the bidirectional state space method and positional embedding to enable the location sensitivity of visual data samples and meet the conditions for global relationship context. An evaluation of the suggested method was carried out by comparison experiments between state-of-the-art algorithms and the proposed methodology. Empirical findings demonstrate that the suggested methodology surpasses the most advanced algorithms on the datasets that are accessible openly. Hence, the suggested approach may be regarded as a helpful tool for therapeutic processes.

**Data Availability Statement:** The dataset RFMiD 2.0 for this study can be found in the website, https://riadd.grand-challenge.org/Download/. The dataset APTOS2019 for this study can be found in the website, https://www.kaggle.com/datasets/

## Introduction

Type 2 diabetes mellitus is a well recognized and significant public health concern, projected to impact over 500 million individuals worldwide by 2045 [1]. The condition has substantial effects on several human organs, such as the eyes, heart, and kidneys. Diabetic retinopathy

mariaherrerot/aptos2019. The RFMiD 2.0 dataset, used in this study, is publicly available and can be accessed without any restrictions. The dataset is hosted on the Grand Challenges in Biomedical Imaging Platform, and interested researchers can download it from the following link: https://riadd. grand-challenge.org/Download/. Readers are advised to create an account on https://riadd. grand-challenge.org/Home/ then access the 'Download' tab https://riadd.grand-challenge.org/ Download/, and the dataset can be downloaded by clicking on 'Training Set', 'Evaluation Set', and 'Test Set' links inside this webpage. And the DOI of this dataset is 10.3390/data6020014. We have also used the public Kaggle dataset in this study. This dataset was considered in the Asia Pacific Tele-Ophthalmology Society (APTOS) 2019 blindness detection competition https://www.kaggle.com/ datasets/mariaherrerot/aptos2019. Readers are advised to create an account on Kaggle. And the dataset folder can be downloaded by clicking on 'Download' button inside this webpage. The dataset Messidor for this study can be found in the website, https://www.adcis.net/en/third-party/ messidor/, and its DOI is 10.5566/ias.1155. A form with personal information needs to be completed to download this dataset. Readers are advised to download this form by clicking on 'Download form' button inside this webpage.

**Funding:** The author(s) received no specific funding for this work.

**Competing interests:** The authors have declared that no competing interests exist.

(DR) is well recognized as a common condition resulting from diabetes [2]. It has the potential to adversely impact the blood vessels in the retina, leading to diminished vision or possibly complete loss of eyesight [3]. Significant proportion of diabetic people would ultimately acquire DR. Moreover, prompt identification and therapy might decelerate the progression of DR. Therefore, the identification and classification of DR in its early stage becomes a crucial job in clinical settings.

Fundus images reveal several structural abnormalities, including as hard exudate, soft exudate, hemorrhage, microaneurysm, and neovascularization, which indicate the development phases of DR. DR lesion identification involves identifying several categories of lesions, whereas DR grading is the assessment of the extent of DR in a retinal image. Both approaches are crucial for the early detection and management of DR in clinical settings. The identification and classification of these abnormalities now depend on manual examination [3–5]. However, manually identifying DR lesions is arduous, time-consuming, and prone to errors. Furthermore, the implementation of manual examination challenges arise from the scarcity of ophthalmologists in the less developed regions. Hence, it is imperative to provide more focus to the automated diagnosis and grading of DR.

Over the past few decades, deep learning methods have demonstrated encouraging results in several use cases, including as image categorization and natural language processing. Numerous deep learning models have been specifically developed for the purpose of detecting and evaluating DR. For example, Rajan and Sreejith [6] introduced the convolutional neural network (CNN) as a method to extract different retinal characteristics such as blood vessels, optic discs, and lesions. This enabled the detection of retinal disorders using fundus images. The present work employed data augmentation techniques, including rotation and transition, to extend the image samples. The study conducted by [7] utilized a CNN model to accurately categorize retinal images as either normal or pathological. The proposed approach started by enhancing the contrast of retinal images to more accurately depict different pathologies. Wang *et al.* [8] employed five distinct CNN architectures for the purpose of disease detection and classification. During the training process, all CNN models were trained to minimize a modified version of binary cross-entropy loss. This paper introduced a hybrid deep learning model, which combined DenseNet [9] and ShuffleNet [10], to address the task of classifying and segmenting retinal blood vessels. The technique proposed by Subramaniam and Naganathan [11] involved the integration of the active gradient deep CNN model with the red spider optimization algorithm. Compared to our method, their work focused on the integration of optimization algorithms, while our approach emphasizes structural innovation of the model and depth of feature extraction. Pandey and Kumar [12] proposed a cascade network using lightweight CNN and CNN Xception network to perform binary classification and multi-grading of DR and diabetic macular edema (DME). Our model can surpass their cascade network in terms of feature extraction and classification accuracy. Recently, Abushawish *et al.* [13] presented a survey of the evolution in deep learning models for DR detection, focusing on the transition from machine learning to deep learning algorithms such as CNNs. Our work complements their research by further enhancing CNNs by integrating vision mamba model to improve the accuracy of DR detection.

CNN-based models can offer robust feature extraction capabilities, operate efficiently due to parameter sharing and sparse connectivity, and exhibit translational invariance, making them highly effective for image processing tasks. However, they have limitations in capturing global image dependencies, may struggle with transformations such as rotation and scaling, require substantial computational resources, and depend heavily on large annotated datasets for training, which can be a challenge in scenarios with limited data availability.

Considerable efforts have been made to develop vision transformer (ViT) models specifically designed for the purpose of retinal image categorization. In their study, Bi *et al.* [14] developed a hybrid framework that combines the global representation capability of ViT [15] with the local representation extraction capability of traditional multiple instance learning (MIL). The implementation of a multiple instance vision transformer (MIL-ViT) involved the independent generation of semantic probability distributions by the vanilla ViT branch and the MIL branch. A bag consistency loss was used to reduce the training error. The study conducted by Halder [16] investigated the capacity of the ViT model to effectively capture complex patterns that were essential for medical image classification and surpassed the performance achieved in benchmark results. Compared to Halder's work, our study not only focuses on model performance but also on the interpretability and practical clinical application of the model. Hemalakshmi *et al.* [17] introduced a hybrid deep learning model that combined SqueezeNet [18] and ViT models. This model leveraged the specific strengths of SqueezeNet and ViT to effectively capture both local and global features of medical images, resulting in precise classification. Our model, while integrating these technologies, also introduces additional innovations such as adaptive feature fusion and multi-scale analysis to further improve classification accuracy. Moreover, in their study, Leite and Danilo introduced a ViT-based model called ViT-BRSET [19], which was designed to identify individuals with heightened excavation of the optic nerve. Our model extends beyond ViT by introducing novel attention mechanisms to improve the recognition of subtle pathological features.

ViT models have emerged as a powerful alternative to CNNs in the field of computer vision, offering several advantages such as the ability to capture long-range dependencies and process inputs of varying sizes, which can enhance their flexibility and potential for greater generalization. They have demonstrated superior performance on standard datasets, showcasing their effectiveness in image classification and other vision tasks. However, ViTs also came with significant challenges, primarily due to their high computational and memory costs associated with the self-attention mechanism, especially when dealing with high-resolution images. This quadratic computing cost can be prohibitive for real-time applications and large-scale deployments. Additionally, ViTs often require substantial labeled data to achieve optimal performance, which can be a limiting factor in certain scenarios.

In this work, we provide a new framework that combines CNN and vision mamba models to effectively detect and grade DR in retinal images. The suggested vision mamba module is notably influenced by the research findings in [20]. The present CNN module is designed to extract the internal structure of each image by utilizing the local receptive field. The suggested vision mamba approach is utilized to achieve global context modeling and location-aware visual recognition. Therefore, the vision mamba module that is being suggested combines both positional embedding and class token. It should be noted that the insertion of the positional token into the suggested architecture enables the provision of position information for each image patch. Moreover, the vision mamba component utilizes the bidirectional state space paradigm (SSM), which incorporates positional awareness through the aforementioned positional token. Although the attention mechanism is not included in the suggested model, our framework offers equivalent capacity to ViT while limiting the computational cost in a sub-quadratic manner. In order to assess the suggested methodology, we performed comparative analyses between cutting-edge techniques and our approach using publically accessible datasets for both lesion categorization and DR quantification. Empirical findings of the suggested approach illustrate the superiority of this study compared to the current leading methods in terms of a range of assessment criteria.

In general, the contributions of this study can be described as follows:

- We introduce a pioneering pipeline that synergizes CNN with a vision mamba for the precise identification of DR lesions. This integration is innovative as it leverages the strengths of CNNs in feature extraction and the efficiency of vision mamba in processing visual data.

- The vision mamba component, which is central to the proposed pipeline, demonstrates a performance parity with ViT in terms of accuracy but with a reduction in computational complexity. This advancement is crucial for applications where resource constraints are prevalent, such as in mobile health diagnostics.

- The experimental findings indicate that the proposed model surpasses existing state-of-the-art algorithms in both the detection and grading of DR. This achievement is pivotal as early and accurate diagnosis is essential for timely intervention and can potentially prevent vision loss among diabetic patients.

To note that the computational complexity of ViT's self-attention mechanism is a quadratic function of the sequence length M, i.e., $4MD^2 + 2M^2D$. In contrast, the computational complexity of the SSM in vision mamba is a linear function of the sequence length M, i.e., $3M(2D)N + M(2D)N$, where N is a fixed parameter, defaulting to 16. This indicates that for long sequences, vision mamba's computational complexity is lower than that of ViT.

## Methodology

### Overall architecture of the proposed model

This work introduced a joint approach of CNN and vision mamba to tackle the tasks of detecting and evaluating DR. The overall structure of the proposed model is illustrated as Fig 1.

As shown in the Fig 1, the process begins with the input image, which is a retinal scan, being fed into the STEM module of the Inception-ResNet-V2 architecture [21]. This module serves as the initial feature extractor, leveraging the strengths of the Inception-ResNet-V2 design to capture a rich set of features from the image. The Inception-ResNet-V2 is a well-established CNN architecture that combines the Inception module's ability to capture features at multiple scales with the ResNet's residual connections to ease the training process and improve performance. Following the STEM module, the feature maps are processed through a series of ResNet-A and ResNet-B blocks, which further refine the features and enhance the model's ability to learn

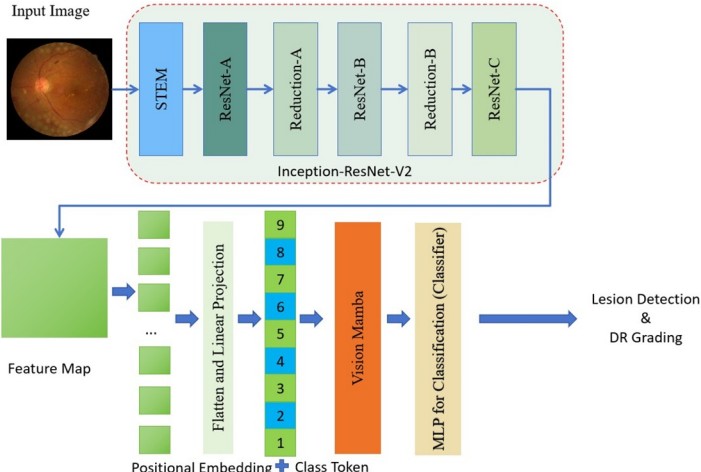

**Fig 1. The pipeline of the proposed CNN-Transformer model.** MLP denotes multi-layer perceptron.

complex patterns. The Reduction-A and Reduction-B layers are then applied to downsample the feature maps, reducing their spatial dimensions while increasing the depth, which is crucial for capturing more abstract representations of the data. The output from the Inception-ResNet-V2 feature extractor is then split into two paths. One path leads to the generation of a feature map, which is a two-dimensional representation of the features extracted from the input image. This feature map is then flattened and linearly projected to create a one-dimensional feature vector. This vector is combined with positional embeddings and a class token, which are critical for providing the model with information about the order of the features and the task of classification, respectively. The other path from the Inception-ResNet-V2 leads directly to the Vision Mamba model. The Vision Mamba is a transformer-based architecture that is designed to process the sequence of features extracted by the CNN. It utilizes self-attention mechanisms to weigh the importance of different features and their relationships within the sequence, allowing the model to focus on the most relevant features for the task at hand. Finally, the processed features from the Vision Mamba are fed into a Multilayer Perceptron (MLP) for classification. The MLP serves as the final classifier, using the features and their relationships to make a decision on the presence and severity of DR lesions in the input image.

This joint approach of CNN and Vision Mamba leverages the strengths of both architectures: the CNN's ability to extract local features and the transformer's capacity to model global dependencies and context. The integration of them is a robust framework for DR lesion detection and grading that has the potential to improve the accuracy and efficiency of automated DR screening systems.

## Stem module

In order to extract features from retinal images, the CNN model Inception-ResNet-V2 employs a sophisticated stem module, as illustrated in Fig 2. This module is designed to efficiently capture a wide range of features from the input images, which is crucial for the subsequent layers to learn complex patterns.

The stem module begins with a standard $3 \times 3$ convolutional layer that serves as the initial feature extractor. This is followed by two additional $3 \times 3$ convolutional layers, which further refine the feature maps by increasing the depth and complexity of the features. The use of multiple $3 \times 3$ convolutions allows the network to capture spatial hierarchies in the data. After the initial convolutional layers, the stem module incorporates a Max-Pooling layer. This operation reduces the spatial dimensions of the feature maps while retaining the most significant features, thus enhancing the model's ability to generalize and reducing the computational load for subsequent layers. The diagram also shows the use of concatenation layers, which combine the

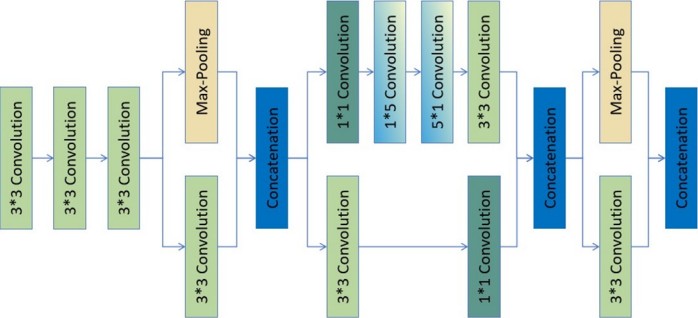

**Fig 2. The stem module in the proposed Inception-Resnet-V2 model.**

feature maps from different branches of the network. This technique allows the model to integrate features extracted at various scales and resolutions, providing a richer representation of the input data. Furthermore, the stem module includes a combination of $1 \times 1$ and $5 \times 5$ convolutional layers. The $1 \times 1$ convolutions are used to linearly combine the features and reduce the dimensionality, while the $5 \times 5$ convolutions enable the model to capture larger receptive fields, which is beneficial for understanding the context within the retinal images. The final concatenation layer in the stem module aggregates the features from all the previous layers, resulting in a comprehensive set of features that are then passed on to the deeper layers of the Inception-ResNet-V2 model. This comprehensive feature set is essential for the model to effectively perform tasks such as detecting and classifying various pathologies in retinal images.

Overall, the stem module of Inception-ResNet-V2 is a critical component that sets the foundation for the model's feature extraction capabilities, enabling it to effectively process and analyze retinal images.

## Vision mamba

This study presents the vision mamba model [20] for the purpose of lesion categorization and DR grading in retinal images. The model is illustrated in Fig 3.

The input of the proposed vision mamba is derived from the output of the Inception-Resnet-v2 model and is partitioned into image patches of linear embeddings, in a sequential manner, as illustrated in Fig 3. The vision mamba is a deep learning architecture that utilizes the SSM mechanism to optimally handle lengthy sequence data, particularly for computer vision applications. Vision mamba enhances the efficiency of image data processing by using the bidirectional SSM architecture, as seen in Fig 3. This architecture provides a notable benefit when handling high-resolution images, in comparison to transformer-based deep learning models. Vision mamba is primarily characterized by its use of positional embeddings to label image sequences and compress visual representations. Furthermore, the architecture of vision mamba draws inspiration from the seminal Kalman filter model [22], known for its ability to capture long-range dependencies and its potential for parallel training. It positions vision mamba as a highly promising substitute for transformers in the domain of machine vision.

## Results

### Dataset

The suggested vision mamba model was intentionally developed utilizing well-known public datasets for grading DR, namely the APTOS2019 database [23] and the Messidor dataset [24].

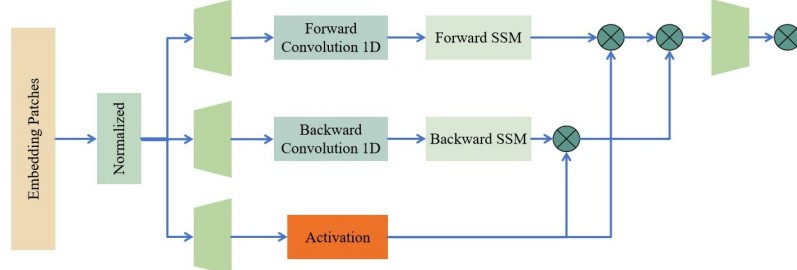

**Fig 3. The structures of the introduced vision mamba model.** SSM denotes state space model introduced in the work of [20].

**Table 1. The distribution of APTOS2019 and Messidor datasets.**

| Class | No. of images in APTOS2019 | No. of images in Messidor |
|---|---|---|
| DR 0 | 1,805 | 546 |
| DR 1 | 370 | 153 |
| DR 2 | 999 | 247 |
| DR 3 | 193 | 254 |
| DR 4 | 295 | - |
| Total | 3,662 | 1,200 |

Each of the 3,662 fundus images in the APTOS2019 database is tagged with one of the five classes of DR, as specified in Table 1. Pursuant to established standards [25–29], we utilized a meticulous 10-fold cross-validation methodology to evaluate the effectiveness of our model on the APTOS2019 dataset.

In contrast, the Messidor dataset consists of 1,200 fundus images that have been categorized with DR grading and DME annotations, as described in Table 1. Using a 10-fold cross-validation technique, we conducted a binary classification job on images categorized as DR grades 0 and 1 from the Messidor dataset, in order to provide a fair comparison with current research [27, 30]. Furthermore, we utilized the whole Messidor dataset to improve our model for the drug resistance grading job, thereby increasing its practicality and resilience in clinical settings.

In this study, we have meticulously partitioned the entire dataset to ensure that the training (80%) and testing set (70%) are devoid of any overlapping patient data, which is a critical step in preventing data leakage. We acknowledge the importance of maintaining the sanctity of our model's evaluation process and have thus implemented stringent measures to preclude the inclusion of the same patient data in both sets. This meticulous exclusion is pivotal in upholding the impartiality of our model's performance assessment and ensures that the generalization capabilities of our model are accurately represented.

To note that there is potential for bias introduced by the imbalance in diabetic retinopathy grades within the used datasets. To mitigate this, we have implemented several strategies to ensure the robustness and fairness of our model's performance, including:

- Class Weight Adjustment: We have incorporated class weight adjustments during model training to counteract the disproportionate representation of classes, thereby enhancing the model's sensitivity to minority classes.

- Comprehensive Evaluation Metrics: We have selected a suite of evaluation metrics that are less sensitive to class imbalance, including precision, recall, F1-score, and AUC-ROC, to provide a balanced assessment of our model's performance across all classes.

- Data Augmentation: To enrich the dataset and reduce the impact of class imbalance, we have applied data augmentation techniques to the minority classes, creating a more diverse and representative training set.

## Implementation details

The suggested model was pre-trained using the extensive ImageNet ISLVRC2012 dataset, which is a comprehensive collection of natural photos available at https://www.image-net.org/challenges/LSVRC/index.php. This dataset consists of more than 120 million photos that

cover 1000 different categories. In order to conform to the specifications of the vision mamba architecture, we adjusted the dimensions of the images from the ImageNet ISLVRC2012 data-set to a consistent resolution of 256 × 256 pixels throughout the pre-training phase. The ini-tialization of our model differs from that of the original ViT described in [31], as we eliminated the use of pre-trained weight parameters. Instead of using the conventional random initialization approach, we choose to initialize the weight parameters of our model based on a particular iteration of the ImageNet dataset [32].

In the pre-training phase of our model, we have strategically formulated a multi-classifica-tion problem, optimizing the training process with a focus on the binary cross-entropy loss function. This loss function, as represented by Eq (1), is pivotal in ensuring that our model is finely tuned to the nuances of natural image data, specifically from the ImageNet ISLVRC2012 dataset. The binary cross-entropy loss is particularly effective for binary classification tasks, where it measures the dissimilarity between the true binary label and the predicted probability, thus driving the model to improve its predictions during training.

To enhance the training process, we have adopted several optimization strategies. Firstly, we have utilized the Inception-ResNet-V2 architecture, excluding the last three layers to adapt it to our specific requirements. This architecture is known for its efficiency in feature extrac-tion, which is crucial for the initial phase of our model's training. Secondly, we have incorpo-rated the Vision Mamba model, which builds upon the features extracted by the Inception-ResNet-V2. The Vision Mamba model, as described in [20], introduces a transformer-based approach that further refines the feature representation and enhances the model's ability to capture complex patterns.

$$Loss(y, y') = \sum_{i=1}^{C} y_i log(y'_i), \tag{1}$$

where $y$ and $y'$ denote the ground-truth label and prediction of the label, respectively.

Additionally, we have implemented a training strategy that includes a careful selection of hyperparameters, such as learning rate, batch size, and the number of epochs. The learning rate determines the step size at each iteration while moving toward a minimum of a loss func-tion, and we have chosen an optimal value that allows for effective convergence without caus-ing divergence. The batch size and the number of epochs were selected based on early stopping to prevent overfitting while ensuring the model trains until convergence. The specific hyper-parameters used in our suggested vision mamba model are thoroughly described in Table 2. In order to handle the changing dimensions of image patches, we employ interpola-tion to estimate the position embeddings for each patch. By providing suitable spatial context, this interpolation approach guarantees that the model can efficiently handle a wide range of

**Table 2. The leveraged hyper-parameters values of the proposed approach.**

| Parameter | Setting |
|---|---|
| Batch size | 8 |
| classes | 5 (DR grading) or 2 (Binary) |
| Image resolution | 256 × 256 |
| optimizer | Adam |
| Learning rate | 1e-6 |
| Patch size | 16 |
| Depth | 12 |
| Epochs | 200 |

image patches, which is crucial for the transformer's attention mechanism to operate well. Equation Eq (1) is the supervisory signal for both the feature extractor and the classification module in the context of retinal image classification, where our model is trained using the cross-entropy loss function. An essential role of this loss function is to direct the model in acquiring discriminative characteristics that are crucial for precise categorization of retinal disorders.

Throughout the training process, a rigorous set of data augmentation methods is used to improve the resilience and applicability of the model. In order to diversity the input data and replicate various views and transformations that the model may experience in real-world circumstances, a range of operations such as rotation, horizontal flipping, and cropping are employed. The PyTorch framework [33] is well recognized for its adaptability and effectiveness in deep learning algorithms, and it greatly facilitates the building and training of the model. The training infrastructure is enhanced by the inclusion of 4 NVIDIA Tesla V100 GPUs, which offer the requisite computing capabilities for supporting the intricate processes associated with neural network training. On average, the model analyzes each image within 420 milliseconds, guaranteeing a prompt and effective execution of the training pipeline.

In the context of our experimental analysis, a comprehensive suite of metrics is meticulously utilized to rigorously assess the efficacy of the model under scrutiny. This evaluation encompasses the Area Under the Receiver Operating Characteristic Curve (AUC-ROC), a pivotal measure that delineates the model's capacity to discern between disparate classes. Concurrently, the metric of Accuracy (Acc) is employed to quantify the ratio of predictions that align with the ground truth. Furthermore, Sensitivity, also known as the True Positive Rate (TPR), is incorporated to appraise the model's proficiency in accurately identifying instances of the positive class. Conversely, Specificity, or the True Negative Rate (TNR), is measured to ascertain the model's aptitude in correctly identifying instances of the negative class. Precision, which is the proportion of true positive predictions relative to all positive predictions, is also considered to evaluate the model's precision in identifying the positive class. Recall, synonymous with Sensitivity, is another metric that gauges the model's effectiveness in capturing all instances of the positive class. Lastly, the F1 score, which is derived as the harmonic mean of Precision and Recall, is deployed as a composite metric that harmoniously integrates the dual aspects of Precision and Recall, thereby providing a balanced assessment of the model's performance.

$$Acc = \frac{TP + TN}{TP + TN + FP + FN}, \tag{2}$$

$$Sensitivity = \frac{TP}{TP + FN}, \tag{3}$$

$$Specificity = \frac{TN}{TN + FP}. \tag{4}$$

$$Precision = \frac{TP}{TP + FP}, \tag{5}$$

$$Recall = \frac{TP}{TP + FN}, \tag{6}$$

$$F1 = 2 * \frac{Precision \times Recall}{Precision + Recall}, \tag{7}$$

The terms TP (True Positive), TN (True Negative), FP (False Positive), and FN (False Negative) have distinct conceptual definitions: The variable TP represents the count of occurrences in which the model accurately predicted the positive class. The variable TN denotes the count of instances in which the model accurately predicted the negative class. On the other hand, FP represents the count of instances in which the model erroneously predicted the positive class when it was actually negative. Lastly, FN represents the count of instances in which the model erroneously predicted the negative class when it was actually positive.

Furthermore, two additional measures are included to enhance comprehension of the model's performance: The Weighted F1 Score (wF1) is a specific form of the F1 score that considers the distribution of the classes. It computes the F1 score for each class and then averages them, taking into account the weight assigned to each class based on the number of instances. This is especially advantageous when working with imbalanced datasets, since it guarantees that the performance on the minority class is not eclipsed by the majority class. Weighted Kappa (wKappa) is a metric of inter-rater agreement that takes into consideration the confounding effect of chance agreement. The weighted Kappa applies this notion to the classification job by quantifying the concordance between the predicted labels and the actual labels, taking into account the class distribution. It is especially beneficial in the field of medical image analysis, as it may offer valuable assessment of the model's predictions in terms of their consistency and dependability.

## Experimental results

**Lesion detection.** In order to juxtapose the efficacy of the proposed methodology with the prevailing deep learning paradigms, a comparative analysis was conducted, encompassing both CNN and transformer-based architectures. This comparative evaluation included a spectrum of models such as U-Net [34], Mask R-CNN [35], ExtremeNet [36], TensorMask [37], Visual Transformer [31], ViT [38], Multi-scale Vision Transformer (MViT) [39], Pyramid Vision Transformer (PVT) [40], Perception Transformer (PiT) [41], and Swin Transformer [42]. The comparative results, as delineated in Table 3, reveal that the proposed approach exhibits a commendable performance when juxtaposed with the extant state-of-the-art techniques.

The comparative analysis was conducted utilizing a diverse array of performance metrics to ensure a holistic and rigorous assessment. The results, as depicted in Table 3, demonstrate that the proposed method has achieved a notably superior performance across a multitude of

**Table 3. Lesion detection comparison between the state-of-the-arts and the proposed approach on the Messidor dataset.**

| Model | AUC | Acc | F1 | Recall | Precision |
|---|---|---|---|---|---|
| U-Net [34] | 0.817 | 0.809 | 0.804 | 0.812 | 0.821 |
| Mask R-CNN [35] | 0.801 | 0.825 | 0.820 | 0.819 | 0.811 |
| ExtremeNet [36] | 0.794 | 0.815 | 0.818 | 0.803 | 0.808 |
| TensorMask [37] | 0.814 | 0.827 | 0.805 | 0.817 | 0.812 |
| Visual Transformer [43] | 0.819 | 0.829 | 0.822 | 0.818 | 0821 |
| ViT [38] | 0.823 | 0.834 | 0.826 | 0.831 | 0.822 |
| MViT [39] | 0.837 | 0.841 | 0.835 | 0.816 | 0.818 |
| PVT [40] | 0.845 | 0.862 | 0.853 | 0.837 | 0.841 |
| PiT [41] | 0.869 | 0.874 | 0.868 | 0.872 | 0.855 |
| Swin Transformer [42] | 0.884 | 0.892 | 0.876 | 0.886 | 0.871 |
| The proposed approach | **0.902** | **0.914** | **0.907** | **0.909** | **0.904** |

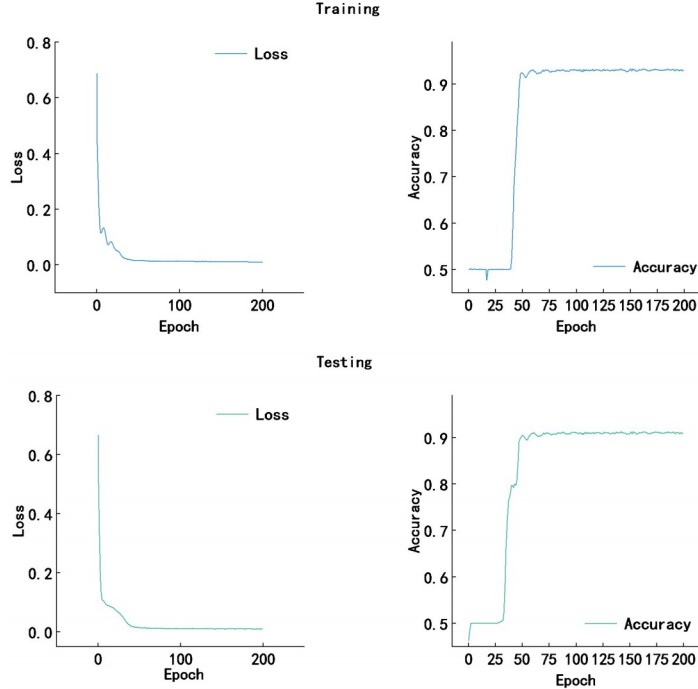

**Fig 4. The loss and accuracy curves of both training and testing processes for the proposed approach on the Messidor dataset.**

evaluation metrics. In particular, it has surpassed current state-of-the-art methodologies with respect to AUC, Acc, F1 Score, Recall, and Precision. These metrics are pivotal in elucidating the classification prowess of the models, offering a nuanced insight into their predictive capabilities. To note that the statistical analysis has revealed that the differences in performance metrics between the proposed approach and the state-of-the-art methods are statistically significant ($p < 0.05$).

Meanwhile, the loss and accuracy curves of both training and testing processes for the proposed approach on the Messidor dataset are provided in Fig 4.

Furthermore, the confusion matrix pertaining to the DR detection task, employing the proposed approach, is presented in Fig 5. This matrix serves as a critical instrument for visualizing the performance of the classification model, particularly in discerning the true positive, true negative, false positive, and false negative predictions, thereby providing a comprehensive overview of the model's diagnostic accuracy.

These results in Fig 5 indicate that the model performs well in identifying both healthy and DR cases, with high accuracy, precision, recall, and F1 scores. However, it is also important to note that despite the model's overall promising performance, a small number of healthy individuals were incorrectly predicted as DR (53 false positives) and a small number of DR individuals were incorrectly predicted as healthy (50 false negatives).

**DR grading.** We performed a set of comparative experiments on the APTOS2019 dataset to evaluate the effectiveness of our method in DR grading. These tests involved comparing our technique with many established methods. The results of these compared trials, as comprehensively described in Table 4, demonstrate that the suggested methodology surpasses the current cutting-edge methods in several assessment criteria. The system we propose has a distinct superiority in the key measures, namely Acc, wF1, and wKappa scores. In the context of DR

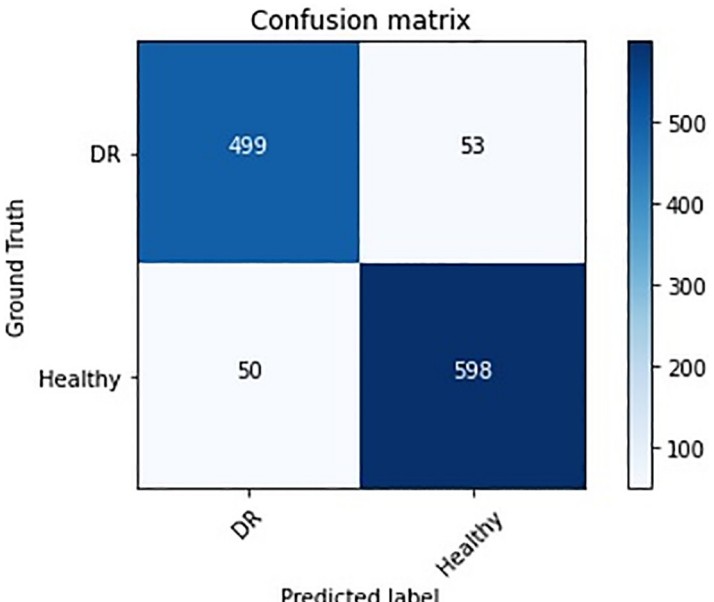

**Fig 5. The confusion matrix for DR and healthy classification on the Messidor dataset.**

grading, these results highlight the resilience and efficacy of our suggested approach. By demonstrating superior performance compared to a wide range of advanced algorithms, this study positions itself as a prominent method for grading DR, providing possible advantages for the early identification and treatment of this common eye disease.

The estimated values of Acc, wF1, and wKappa obtained by the suggested method demonstrate that our model is very efficient in evaluating DR, even when there is an imbalance in class distribution. An imbalance in class distribution can frequently result in a model that exhibits bias towards the dominant class, therefore leading to worse performance for the minority classes. Nevertheless, the performance of our model indicates its ability to precisely assess both prevalent and uncommon DR grades, a critical aspect for timely identification and suitable clinical intervention. One notable feature of our proposed method is its capability to effectively address unbalanced classification problems, which are frequently encountered in

**Table 4. DR grading comparison between the state-of-the-arts and the proposed approach.**

| Model | AUC | Acc | wF1 | wKappa |
|---|---|---|---|---|
| U-Net [34] | 0.805 | 0.814 | 0.823 | 0.817 |
| Mask R-CNN [35] | 0.822 | 0.836 | 0.845 | 0.831 |
| ExtremeNet [36] | 0.854 | 0.848 | 0.842 | 0.857 |
| TensorMask [37] | 0.871 | 0.852 | 0.851 | 0.843 |
| Visual Transformer [43] | 0.903 | 0.856 | 0.859 | 0.862 |
| ViT [38] | 0.911 | 0.876 | 0.869 | 0.878 |
| MViT [39] | 0.933 | 0.882 | 0.874 | 0.895 |
| PVT [40] | 0.958 | 0.891 | 0.897 | 0.901 |
| PiT [41] | 0.971 | 0.897 | 0.904 | 0.908 |
| Swin Transformer [42] | 0.982 | 0.902 | 0.894 | 0.917 |
| The proposed approach | **0.985** | **0.906** | **0.903** | **0.926** |

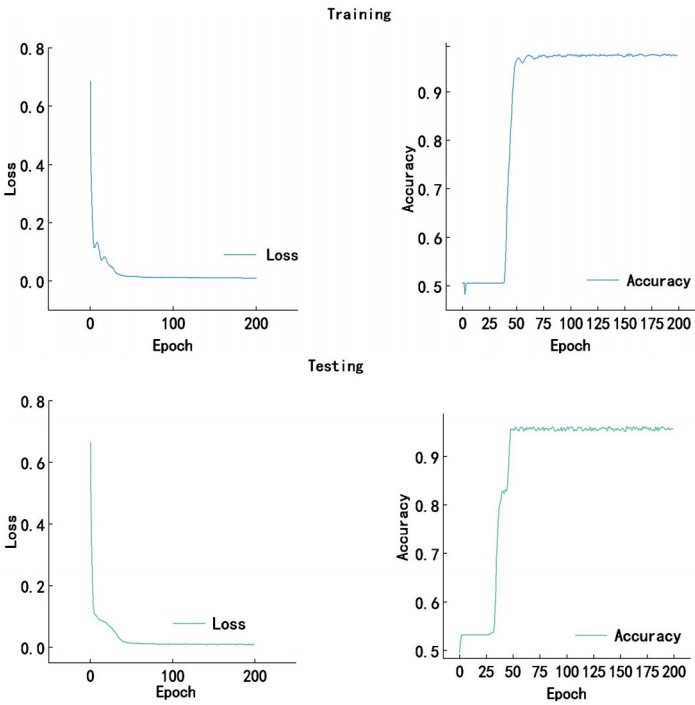

**Fig 6. The loss and accuracy curves of both training and testing processes for the proposed approach on the APTOS2019 dataset.**

real-world applications such as medical image analysis. Our model showcases its potential relevance in practical applications by delivering precise and dependable predictions for all grades of DR. This enables better-informed therapeutic decisions and enhanced patient outcomes. To note that the statistical analysis has revealed that the differences in performance metrics between the proposed approach and the state-of-the-art methods are statistically significant ($p < 0.05$).

Meanwhile, the loss and accuracy curves of both training and testing processes for the proposed approach on the APTOS2019 dataset are provided in Fig 6.

The training and testing curves for the proposed DR grading approach exhibit a rapid decline in loss and a corresponding increase in accuracy, both stabilizing after approximately 100 epochs. The training curves indicate a model that has converged with high accuracy, while the testing curves, although slightly higher in loss and slightly lower in accuracy, demonstrate the model's robust generalization capabilities. This suggests that while the model performs exceptionally well on the training set, it also maintains a high level of accuracy on the test set, albeit with a hint of overfitting as indicated by the slight discrepancy between the two sets' performance metrics.

Furthermore, the confusion matrix for the DR grading job calculated using the suggested method can be seen in Fig 7.

The confusion matrix as shown in Fig 7 shows the number of instances for each actual DR grade (Ground Truth) that were predicted to be each of the possible grades (Predicted label):

DR0: The model correctly predicted 1717 instances as DR0 (True Negatives for DR0). However, it misclassified 30 instances of DR1, 27 of DR2, 18 of DR3, and 13 of DR4 as DR0 (False Negatives for those grades, respectively).

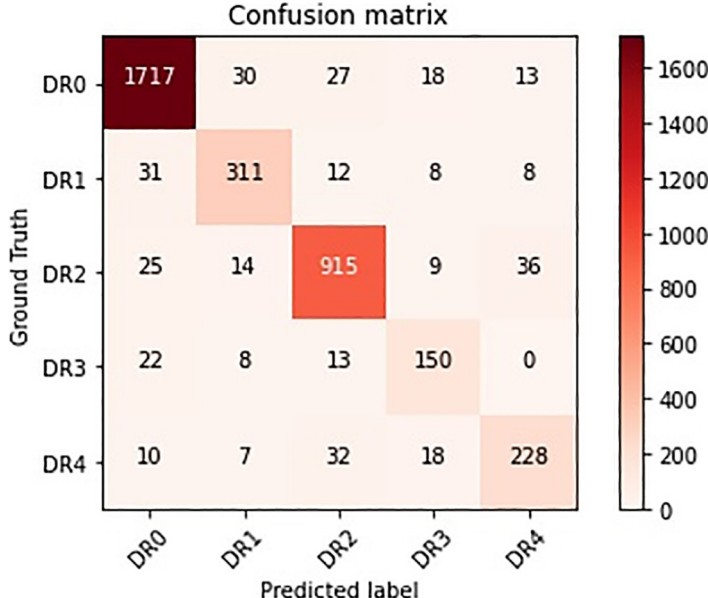

**Fig 7. The confusion matrix for DR grading on the APTOS2019 dataset by using the proposed approach.**

DR1: For DR1, the model correctly predicted 311 instances (True Positives for DR1). It had 31 misclassifications from DR0, 12 from DR2, 8 from DR3, and 8 from DR4.
DR2: The model accurately predicted 915 instances as DR2 (True Positives for DR2), which is a strong performance for this grade. It had 25 misclassifications from DR0, 14 from DR1, 9 from DR3, and 36 from DR4.

DR3: For DR3, the model correctly predicted 150 instances (True Positives for DR3). It had 22 misclassifications from DR0, 8 from DR1, 13 from DR2, and none from DR4.

DR4: The model correctly predicted 228 instances as DR4 (True Positives for DR4). It had 10 misclassifications from DR0, 7 from DR1, 32 from DR2, and 18 from DR3.

Moreover, we demonstrate the fundus images from the datasets used by the proposed approach. The images showcase the progression from DR 0, DR1, DR 2, DR 3, and DR 4 (as shown in Fig 8). Each image has been meticulously selected to represent the typical cases of DR grading within the datasets. These images not only exhibit the capability of the CNN-Vision Mamba model in identifying and classifying different levels of DR but also visually attest to the model's accuracy and reliability when dealing with actual fundus images.

## Ablation study

The performance of the suggested technique was evaluated by an ablation study, which involved pre-training on either ImageNet ISLVRC or a combination of ImageNet ISLVRC and RFMiD. Furthermore, we assessed the suggested model using various combinations of the embedding dimension for the newly presented vision mamba. It should be noted that the ablation experiments were conducted using the APTOS2019 and Messidor datasets. Our chosen metrics for the multi-classification DR grading job in the APTOS2019 dataset were AUC, Acc, wF1, and wKappa. To assess the performance of DR detection in the Messidor dataset, we employed the metrics of AUC, Acc, F1 score, Recall, and Precision. The results of the ablation experiments are displayed in Tables 5 and 6.

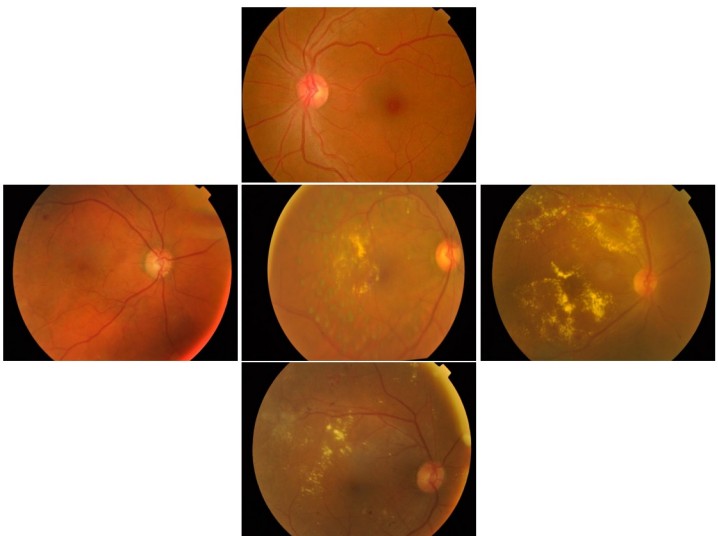

**Fig 8. The fundus images in the dataset using by the proposed approach.** (Top) DR 0; (Middle Left) DR1; (Center) DR 2; (Middle Right) DR 3; (Bottom) DR 5.

**Table 5. Outcome of the ablation study on the Messidor dataset.**

| Combination | | DR Detection | | | | |
|---|---|---|---|---|---|---|
| Pre-train | Heads | AUC | Acc | F1 | Recall | Precision |
| ImageNet ISLVRC | 128 | 0.875 | 0.872 | 0.886 | 0.878 | 0.891 |
| ImageNet ISLVRC | 256 | 0.879 | 0.889 | 0.892 | 0.880 | 0.899 |
| ImageNet+RFMiD | 128 | 0.897 | 0.905 | 0.901 | 0.898 | 0.908 |
| ImageNet+RFMiD | 256 | 0.902 | 0.914 | 0.911 | 0.908 | **0.912** |

**Table 6. Outcome of the ablation study on the APTOS2019 dataset.**

| Combination | | DR Grading | | | |
|---|---|---|---|---|---|
| Pre-train | Heads | AUC | Acc | wF1 | wKappa |
| ImageNet ISLVRC | 128 | 0.916 | 0.883 | 0.885 | 0.905 |
| ImageNet ISLVRC | 256 | 0.938 | 0.891 | 0.892 | 0.916 |
| ImageNet +RFMiD | 128 | 0.959 | 0.903 | 0.898 | 0.918 |
| ImageNet+RFMiD | 256 | 0.985 | 0.907 | 0.903 | 0.926 |

**Discussion.** The present work underscores the importance of the class token in vision algorithms, which traditionally may overlook the wealth of information embedded in individual image patches. Each patch, rich in relative information, holds the potential to enhance classification accuracy when adequately leveraged. In the context of clinical diagnostics, where lesions can be ubiquitous within retinal images, the feature representation of the class token in conventional ViTs could be enhanced by incorporating contextual information from these image patches.

Our findings reveal that the positional information preserved in each image patch is crucial, especially considering the origins of transformers in sequential data processing. This positional

awareness allows our model to maintain the global context of the image, which is paramount for accurate lesion detection in medical imaging. In addition, the incorporation of softmax and pooling operations into the self-attention module, as suggested, is not only pivotal for unveiling the global receptive field but also for optimizing computational efficiency. This innovative approach addresses the trade-off between the expressive capacity required for complex pattern recognition and the computational complexity inherent in the attention mechanism, which is particularly relevant in resource-constrained clinical settings. Empirical results have demonstrated that our proposed CNN-Vision Mamba model excels in capturing feature embeddings and reliably detecting global interconnections within retinal images. This capability is transformative for the comprehensive mapping of features, which is indispensable for the identification of abnormalities. The simplified model framework presented in this work is not only compatible with various ViT variants but also offers a modular integration option, enhancing its applicability and flexibility in diverse clinical scenarios.

However, we acknowledge the inherent limitations of this study. The performance of deep learning models is influenced by the dataset size, with the number of images being directly proportional to model outcomes. Therefore, expanding the dataset could potentially enhance model generalization and robustness. Additionally, our evaluation criteria, while standard, may benefit from a broader scope. Incorporating a wider range of variables could provide a more nuanced assessment of model performance, particularly in capturing the subtleties of retinal pathologies. Finally, we acknowledge that while the proposed CNN-Vision Mamba model demonstrates excellence in capturing feature embeddings and detecting global interconnections within retinal images, there are limitations that must be considered for real-world deployment. For instance, the model's reliance on high-quality, positional information may be challenged by variations in image acquisition across different clinical settings. Additionally, while we have optimized for computational efficiency, the practical constraints of clinical environments, such as the availability of processing power and memory, may impact the model's scalability and speed of execution.

## Conclusion

In addressing the intricate challenge of classifying retinal imagery, this study introduces an innovative attention mechanism seamlessly integrated into the conventional ViT framework. The research endeavor focuses on the exploration of the synergistic potential of linear and softmax attention modules within the context of retinal lesion identification. While CNNs have garnered considerable acclaim for their prowess in image classification by unearthing intrinsic properties of images through the emphasis on local features, they are inherently limited in their ability to capture the broader contextual information. Conversely, ViTs, by leveraging feature embeddings derived from CNNs, are adept at encapsulating the global contextual nuances of the imagery. The proposed hybrid model, following a pre-training regimen on a vast corpus of natural images and subsequent fine-tuning with specialized retinal image datasets, has evinced a remarkable proficiency in the detection of retinal lesions. This model has demonstrated a superior performance when compared to the prevailing CNNs and ViTs, thereby underscoring its efficacy in the domain of retinal image analysis.

As we chart the course for future research, we are poised to delve into the realm of diverse backbone networks as potential feature extractors, and to scrutinize an array of classification algorithms. Given the auspicious outcomes garnered in the realm of retinal image analysis, the ambition is to extend the applicability of the proposed architectural paradigm to encompass a wider spectrum of image classification tasks. This expansion is anticipated to further validate the versatility and robustness of the model in diverse imaging contexts.

## Author Contributions

**Formal analysis:** Zenglei Liu, Ailian Gao, Hui Sheng, Xueling Wang.

**Investigation:** Zenglei Liu, Ailian Gao, Hui Sheng, Xueling Wang.

**Methodology:** Hui Sheng, Xueling Wang.

**Project administration:** Ailian Gao, Xueling Wang.

**Software:** Hui Sheng.

**Supervision:** Xueling Wang.

**Validation:** Zenglei Liu, Ailian Gao.

**Visualization:** Hui Sheng.

**Writing – original draft:** Hui Sheng, Xueling Wang.

**Writing – review & editing:** Zenglei Liu, Ailian Gao.

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
