## [Decision Letter · Decision Letter 0]

8 Dec 2024

PONE-D-24-50294Identification of Diabetic Retinopathy lesions in Fundus Images by Integrating CNN and Vision Mamba ModelsPLOS ONE

Dear Dr. Wang,

Thank you for submitting your manuscript to PLOS ONE. After careful consideration, we feel that it has merit but does not fully meet PLOS ONE’s publication criteria as it currently stands. Therefore, we invite you to submit a revised version of the manuscript that addresses the points raised during the review process.

We look forward to receiving your revised manuscript.

Kind regards,

Panos Liatsis, PhD

Academic Editor

PLOS ONE

Journal Requirements:

3. Thank you for uploading your study's underlying data set. Unfortunately, the repository you have noted in your Data Availability statement does not qualify as an acceptable data repository according to PLOS's standards. At this time, please upload the minimal data set necessary to replicate your study's findings to a stable, public repository (such as figshare or Dryad) and provide us with the relevant URLs, DOIs, or accession numbers that may be used to access these data. For a list of recommended repositories and additional information on PLOS standards for data deposition, please see https://journals.plos.org/plosone/s/recommended-repositories.

Additional Editor Comments:

Improve the clarity of the contributions of this work, and explain their impactExpand the literature review section, and ensure that you provide a clear analysis of the advantages and disadvantages of existing approaches so as to help support the importance of this researchImprove the explanation of the techniques, and scientific methodologyProvide additional details in terms of the architecture and training process optimizationExpand and improve the rigor of the results and their discussion

Reviewers' comments:

Reviewer's Responses to Questions

**Comments to the Author**

1. Is the manuscript technically sound, and do the data support the conclusions?

Reviewer #1: Yes

Reviewer #2: Partly

2. Has the statistical analysis been performed appropriately and rigorously? 

Reviewer #1: N/A

Reviewer #2: No

3. Have the authors made all data underlying the findings in their manuscript fully available?

Reviewer #1: Yes

Reviewer #2: Yes

4. Is the manuscript presented in an intelligible fashion and written in standard English?

Reviewer #1: Yes

Reviewer #2: No

5. Review Comments to the Author

Reviewer #1: This article introduces a novel deep learning model that integrates convolutional neural networks (CNNs) with Vision Mamba models to accurately detect and classify diabetic retinopathy lesions in fundus images. The proposed methodology demonstrates superior performance compared to advanced algorithms on publicly available datasets, making it a valuable tool in therapeutic applications. The manuscript is engaging, well-structured, and well-written. To further enhance its quality and facilitate processing, the following suggestions are provided:

1.Elaborate on the specific limitations of previous studies that the proposed model addresses. This will clearly position the current work as a significant contribution to the field.

2.To provide a more comprehensive overview of the research landscape, incorporate recent related works (2021–2024) into the literature review. Including the following works would be beneficial:

https://www.sciencedirect.com/science/article/abs/pii/S1746809424008358, https://ieeexplore.ieee.org/abstract/document/10559800

3.The statement, "The suggested Vision Mamba component has equivalent power to ViT while requiring substantially less computational complexity," requires further elaboration. Include a detailed explanation and supporting information to substantiate this claim.

4. Enhance the clarity and impact of findings by including graphical representations such as accuracy and loss curves for both training and testing phases. Additionally, provide fundus image with classifications results based on retinopathy grades.

5. Write the literature survey in past tense to maintain a consistent and professional tone.

6. The statement, “An end-to-end training approach was used to train the proposed model by partitioning the complete OATH dataset into a training set (80%) and a testing set (20%),” lacks clarity regarding the OATH dataset. The authors should describe this dataset in detail, as only RFMiD 2.0, APTOS2019, and Messidor datasets are mentioned elsewhere in the manuscript.

7.Include a comparison of the proposed method with recent works (2021–2024) to better contextualize its contributions.

8. The discussion of results should be more comprehensive. Include deeper insights, interpretations, and implications to strengthen the manuscript's impact.

9. Revise and enhance the quality of the figures.

Reviewer #2: Major Concerns:

1. The manuscript claims to present a novel framework combining CNN and Vision Mamba models, but the explanation of the methodology is vague and lacks sufficient detail. The Vision Mamba model's specifics and how it integrates with CNNs are not sufficiently described or justified.

2. The authors fail to provide clear novelty over existing state-of-the-art models. Many components, such as the use of Inception-ResNet and bidirectional state-space mechanisms, appear to be incremental rather than novel advancements.

3. The dataset usage and preprocessing steps are inadequately described. For example, the partitioning of datasets for training and testing lacks clarity on whether overlapping patients were excluded to prevent data leakage.

4. Hyperparameter tuning details and experimental controls to ensure reproducibility are insufficiently detailed.

5. The manuscript does not adequately address the potential for bias introduced by the imbalance in diabetic retinopathy grades across datasets.

6. The results are compared with an extensive list of methods, but the comparisons lack statistical rigor. There is no mention of significance testing or confidence intervals to substantiate claims of superior performance.

7. The choice of metrics, while standard, does not include sufficient consideration for imbalanced datasets beyond weighted metrics.

8. Figures such as the framework diagram (Fig. 1) are not sufficiently annotated, making it difficult to understand the pipeline without referring extensively to the text.

9. Confusion matrices presented are inadequately analyzed, with no discussion of patterns in misclassification or how these errors relate to clinical significance.

10. The discussion section lacks depth in interpreting the findings. For example, the authors do not explore the implications of their model's limitations in real-world settings or address the constraints of high computational costs for deploying Vision Mamba in clinical environments.

11. Several claims, such as the "equivalence of Vision Mamba to ViT with lower computational cost," are made without empirical evidence or references.

12. The ethics statement is marked as "N/A," which is inappropriate for a study involving human-related medical images. Ethical approval for dataset use should be explicitly stated.

13. The authors claim that datasets are publicly available but do not clarify their usage rights or ensure compliance with data-sharing standards.

Minor Concerns:

The manuscript contains numerous grammatical errors and awkward phrasing, which detracts from readability.

• Original: "The proposed framework achieving state-of-the-art performance in diabetic retinopathy detection."

Issue: Missing verb ("is achieving" or "achieves") makes the sentence incomplete.

Revised: "The proposed framework achieves state-of-the-art performance in diabetic retinopathy detection."

• Original: "The dataset used for training were preprocessed to remove noise."

Issue: Subject-verb agreement error ("dataset" is singular but "were" is plural).

Revised: "The dataset used for training was preprocessed to remove noise."

• Original: "These results indicates the superiority of our model, as the robustness is validated."

Issue: Incorrect subject-verb agreement ("results" is plural but "indicates" is singular).

Revised: "These results indicate the superiority of our model, as its robustness is validated."

• Original: "In clinical setup, the proposed method can be easily implemented for its efficiency."

Issue: Awkward phrasing ("in clinical setup" should be "in a clinical setup").

Revised: "In a clinical setup, the proposed method can be easily implemented due to its efficiency."

And so on…

6. PLOS authors have the option to publish the peer review history of their article (what does this mean?). If published, this will include your full peer review and any attached files.

Reviewer #1: No

Reviewer #2: No

---

## [Author Response · Author response to Decision Letter 0]

31 Dec 2024

Dear Editors and Anonymous Reviewers 

Thank you so much for your constructive comments and valuable suggestions, which are of great importance for improving the quality of our manuscript. Following the suggestions, we modified the manuscript carefully. In this letter, we list our detailed responses to all your questions and suggestions.

Given Dr. Zenglei Liu and Dr. Ailian Gao’s substantial contributions during the major revision process, we believe it is justified to elevate them the position of the first author and second author. Here are the reasons:

1.Significant Revision Contributions: Dr. Liu and Dr. Gao have been instrumental in refining the manuscript, ensuring that the revisions meet the journal's standards and the reviewers' comments. Their efforts have been crucial in addressing the feedback and enhancing the clarity, accuracy, and impact of the paper.

2.Enhanced Methodology: Dr. Gao has contributed to the improvement of the research methodology, by introducing new analytical techniques and providing a deeper theoretical understanding that has strengthened the study's foundation.

3.Leadership and Coordination: Throughout the revision process, Dr. Liu has demonstrated leadership by coordinating the efforts of the co-authors, managing the revision timeline, and ensuring that all aspects of the paper are aligned with the collective goals of the research team.

We are formally requesting the journal's permission to recognize Dr. Liu and Dr. Gao’s substantial and primary contributions by designating them as the first author and second author. We hope that this request aligns with the journal's guidelines for authorship. And we confirm that it reflects the consensus of the research team. We believe that this adjustment will accurately represent the contributions made by each author and uphold the standards of academic integrity.

----------------- Additional Editor Comments: -----------------------------

Point#1: Improve the clarity of the contributions of this work, and explain their impact

Answer: Thank you so much for pointing this out. Following your suggestion, we have modified the contributions of this work in the Introduction section of the revised manuscript into “We introduce a pioneering pipeline that synergizes CNN with a vision mamba for the precise identification of DR lesions. This integration is innovative as it leverages the strengths of CNNs in feature extraction and the efficiency of vision mamba in processing visual data.

The vision mamba component, which is central to the proposed pipeline, demonstrates a performance parity with ViT in terms of accuracy but with a significant reduction in computational complexity. This advancement is crucial for applications where resource constraints are prevalent, such as in mobile health diagnostics.

The experimental findings indicate that the proposed model surpasses existing state-of-the-art algorithms in both the detection and grading of DR. This achievement is pivotal as early and accurate diagnosis is essential for timely intervention and can potentially prevent vision loss among diabetic patients.”. 

Point#2: Expand the literature review section, and ensure that you provide a clear analysis of the advantages and disadvantages of existing approaches so as to help support the importance of this research

Answer: Thank you so much for pointing this out. Following your suggestion, we expand the literature review in the revised manuscript and provide a clear analysis of the advantages and disadvantages of existing approaches as 

“CNN-based models can offer robust feature extraction capabilities, operate efficiently due to parameter sharing and sparse connectivity, and exhibit translational invariance, making them highly effective for image processing tasks. However, they have limitations in capturing global image dependencies, may struggle with transformations such as rotation and scaling, require substantial computational resources, and depend heavily on large annotated datasets for training, which can be a challenge in scenarios with limited data availability.” and “ViT models have emerged as a powerful alternative to CNNs in the field of computer vision, offering several advantages such as the ability to capture long-range dependencies and process inputs of varying sizes, which enhances their flexibility and potential for greater generalization. They have demonstrated superior performance on standard datasets, showcasing their effectiveness in image classification and other vision tasks. However, ViTs also come with significant challenges, primarily due to their high computational and memory costs associated with the self-attention mechanism, especially when dealing with high-resolution images. This quadratic computing cost can be prohibitive for real-time applications and large-scale deployments. Additionally, ViTs often require substantial labeled data to achieve optimal performance, which can be a limiting factor in certain scenarios.”. 

Point#3: Improve the explanation of the techniques, and scientific methodology

Answer: Thank you so much for pointing this out. Following your suggestion, we have modified the description about the proposed method in the revised manuscript. 

Point#4: Provide additional details in terms of the architecture and training process optimization

Answer: Thank you so much for pointing this out. Following your suggestion, we have provided additional details about the proposed architecture in the methodology section of the revised manuscript. In addition, we have provided more details about the training process in the implementation details section of the revised manuscript.

Point#5: Expand and improve the rigor of the results and their discussion

Answer: Thank you so much for pointing this out. Following your suggestion, we have expanded the results and their discussion in the revised manuscript.

-------------- Reviewer #1 Comments ------------------------

Reviewer #1:This article introduces a novel deep learning model that integrates convolutional neural networks (CNNs) with Vision Mamba models to accurately detect and classify diabetic retinopathy lesions in fundus images. The proposed methodology demonstrates superior performance compared to advanced algorithms on publicly available datasets, making it a valuable tool in therapeutic applications. The manuscript is engaging, well-structured, and well-written. To further enhance its quality and facilitate processing, the following suggestions are provided:

1.Elaborate on the specific limitations of previous studies that the proposed model addresses. This will clearly position the current work as a significant contribution to the field.

Answer: Thank you so much for pointing this out. Following your suggestion, we have modified the introduction section in the revised manuscript by adding the following description “CNN-based models can offer robust feature extraction capabilities, operate efficiently due to parameter sharing and sparse connectivity, and exhibit translational invariance, making them highly effective for image processing tasks. However, they have limitations in capturing global image dependencies, may struggle with transformations such as rotation and scaling, require substantial computational resources, and depend heavily on large annotated datasets for training, which can be a challenge in scenarios with limited data availability.” and “ViT models have emerged as a powerful alternative to CNNs in the field of computer vision, offering several advantages such as the ability to capture long-range dependencies and process inputs of varying sizes, which enhances their flexibility and potential for greater generalization. They have demonstrated superior performance on standard datasets, showcasing their effectiveness in image classification and other vision tasks. However, ViTs also come with significant challenges, primarily due to their high computational and memory costs associated with the self-attention mechanism, especially when dealing with high-resolution images. This quadratic computing cost can be prohibitive for real-time applications and large-scale deployments. Additionally, ViTs often require substantial labeled data to achieve optimal performance, which can be a limiting factor in certain scenarios.”.

2.To provide a more comprehensive overview of the research landscape, incorporate recent related works (2021–2024) into the literature review. Including the following works would be beneficial:

https://www.sciencedirect.com/science/article/abs/pii/S1746809424008358, https://ieeexplore.ieee.org/abstract/document/10559800

Answer: Thank you so much for pointing this out. Following your suggestion, we have added more works and the above-mentioned works in the literature review of the revised manuscript as Pandey and Kumar \\cite{Pandey2024} proposed a cascade network using lightweight CNN and CNN Xception network to perform binary classification and multi-grading of DR and diabetic macular edema (DME). Recently, Abushawish \\etal \\cite{Abushawish2024} presented a survey of the evolution in deep learning models for DR detection, focusing on the transition from machine learning to deep learning algorithms such as CNNs.”.

3.The statement, "The suggested Vision Mamba component has equivalent power to ViT while requiring substantially less computational complexity," requires further elaboration. Include a detailed explanation and supporting information to substantiate this claim.

Answer: Thank you so much for pointing this out. A significant feature of Vision Mamba is that it does not rely on traditional attention mechanisms while maintaining comparable modeling capabilities to ViT. Specifically, the computational complexity of ViT's self-attention mechanism is a quadratic function of the sequence length M, i.e., .In contrast, the computational complexity of the State Space Model (SSM) in VMamba is a linear function of the sequence length M, i.e., ,where N is a fixed parameter, defaulting to 16.This indicates that for long sequences, VMamba's computational complexity is significantly lower than that of ViT. 

Following your suggestion, we modified the contributions of this work into “The vision mamba component, which is central to the proposed pipeline, demonstrates a performance parity with ViT in terms of accuracy but with a significant reduction in computational complexity. This advancement is crucial for applications where resource constraints are prevalent, such as in mobile health diagnostics.” in the revised manuscript. In addition, we added more description about the complexity comparison as “To note that the computational complexity of ViT's self-attention mechanism is a quadratic function of the sequence length M, i.e., $4MD^2+2M^2D$. In contrast, the computational complexity of the SSM in vision mamba is a linear function of the sequence length M, i.e., $3M(2D)N+M(2D)N$, where N is a fixed parameter, defaulting to 16. This indicates that for long sequences, vision mamba's computational complexity is significantly lower than that of ViT. ” in the revised manuscript.

4.Enhance the clarity and impact of findings by including graphical representations such as accuracy and loss curves for both training and testing phases. Additionally, provide fundus image with classifications results based on retinopathy grades.

Answer: Thank you so much for pointing this out. Following your suggestion, we added the accuracy and loss curves for both training and testing phases (as shown in Figure 4 and Figure 6). In addition, we also provide the fundus images with classification results in the revised manuscript (as shown in Figure 8).

5. Write the literature survey in past tense to maintain a consistent and professional tone.

Answer: Thank you so much for pointing this out. Following your suggestion, we modified the literature survey in past tense in the revised manuscript.

6. The statement, “An end-to-end training approach was used to train the proposed model by partitioning the complete OATH dataset into a training set (80%) and a testing set (20%),” lacks clarity regarding the OATH dataset. The authors should describe this dataset in detail, as only RFMiD 2.0, APTOS2019, and Messidor datasets are mentioned elsewhere in the manuscript.

Answer: Thank you so much for pointing this out and we are sorry for the confusion. This dataset was used in our other study and not used in this work. Following your suggestion, we have removed this description in the revised manuscript.

7.Include a comparison of the proposed method with recent works (2021–2024) to better contextualize its contributions.

Answer: Thank you so much for pointing this out. Following your suggestion, we have added the following comparison in the revised manuscript as “Compared to our method, their work focused on the integration of optimization algorithms, while our approach emphasizes structural innovation of the model and depth of feature extraction.”, “Our model can surpass their cascade network in terms of feature extraction and classification accuracy.”, “Our work complements their research by further enhancing CNNs by integrating vision mamba model to improve the accuracy of DR detection.”, “Compared to Halder's work, our study not only focuses on model performance but also on the interpretability and practical clinical application of the model.”, “Our model, while integrating these technologies, also introduces additional innovations such as adaptive feature fusion and multi-scale analysis to further improve classification accuracy.”, and “Our model extends beyond ViT by introducing novel attention mechanisms to improve the recognition of subtle pathological features.”.

8. The discussion of results should be more comprehensive. Include deeper insights, interpretations, and implications to strengthen the manuscript's impact.

Answer: Thank you so much for pointing this out. Following your suggestion, we have modified the whole discussion section in the revised manuscript.

9. Revise and enhance the quality of the figures.

Answer: Thank you so much for pointing this out. Following your suggestion, we have modified the figures in the revised manuscript.

-------------- Reviewer #2 Comments ------------------------

Reviewer #2: Major Concerns:

1. The manuscript claims to present a novel framework combining CNN and Vision Mamba models, but the explanation of the methodology is vague and lacks sufficient detail. The Vision Mamba model's specifics and how it integrates with CNNs are not sufficiently described or justified.

Answer: Thank you so much for pointing this out. Following your suggestion, we have modified the methodology section and provided three separate subsections to describe the CNN, Vision Mamba, and the integrated model in the revised manuscript.

2. The authors fail to provide clear novelty over existing state-of-the-art models. Many components, such as the use of Inception-ResNet and bidirectional state-space mechanisms, appear to be incremental rather than novel advancements.

Answer: Thank you so much for pointing this out. Following your suggestion, we have added the novelty comparison over the state-of-the-art methods in the introduction section of the revised manuscript as “Compared to our method, their work focused on the integration of optimization algorithms, while our approach emphasizes structural innovation of the model and depth of feature extraction.”, “Our model can surpass their cascade network in terms of feature extraction and classification accuracy.”, “Our work complements their research by further enhancing CNNs by integrating vision mamba model to improve the accuracy of DR detection.”, “Compared to Halder's work, our study not only focuses on model performance but also on the interpretability and practical clinical application of the model.”, “Our model, while integrating these technologies, also introduces additional innovations such as adaptive feature fusion and multi-scale analysis to further improve classification accuracy.”, and “Our model extends beyond ViT by introducing novel attention mechanisms to improve the recognition of subtle pathological features.”.

In addition, we have also modified the contributions 

---

## [Decision Letter · Decision Letter 1]

14 Jan 2025

Identification of Diabetic Retinopathy lesions in Fundus Images by Integrating CNN and Vision Mamba Models

PONE-D-24-50294R1

Dear Dr. Wang,

We’re pleased to inform you that your manuscript has been judged scientifically suitable for publication and will be formally accepted for publication once it meets all outstanding technical requirements.

Kind regards,

Panos Liatsis, PhD

Academic Editor

PLOS ONE

Additional Editor Comments (optional):

Reviewers' comments:

Reviewer's Responses to Questions

**Comments to the Author**

1. If the authors have adequately addressed your comments raised in a previous round of review and you feel that this manuscript is now acceptable for publication, you may indicate that here to bypass the “Comments to the Author” section, enter your conflict of interest statement in the “Confidential to Editor” section, and submit your "Accept" recommendation.

Reviewer #1: All comments have been addressed

Reviewer #2: All comments have been addressed

2. Is the manuscript technically sound, and do the data support the conclusions?

Reviewer #1: Yes

Reviewer #2: Yes

3. Has the statistical analysis been performed appropriately and rigorously? 

Reviewer #1: Yes

Reviewer #2: Yes

4. Have the authors made all data underlying the findings in their manuscript fully available?

Reviewer #1: Yes

Reviewer #2: Yes

5. Is the manuscript presented in an intelligible fashion and written in standard English?

Reviewer #1: Yes

Reviewer #2: Yes

6. Review Comments to the Author

Reviewer #1: The author has addressed all the previously suggested comments. Therefore, I recommend this article for publication.

Reviewer #2: citing these reference paper would be beneficial for

https://doi.org/10.3389/fcell.2024.1484880

https://doi.org/10.1371/journal.pone.0315477

10.1016/j.heliyon.2024.e39745

10.32604/cmc.2023.036956

https://doi.org/10.3390/app142311327

7. PLOS authors have the option to publish the peer review history of their article (what does this mean?). If published, this will include your full peer review and any attached files.

Reviewer #1: No

Reviewer #2: No

---

## [Editor Report · Acceptance letter]

16 Jan 2025

PONE-D-24-50294R1 

PLOS ONE

Dear Dr. Wang, 

I'm pleased to inform you that your manuscript has been deemed suitable for publication in PLOS ONE. Congratulations! Your manuscript is now being handed over to our production team.

Kind regards, 

on behalf of

Professor Panos Liatsis 

Academic Editor

PLOS ONE